# Validation of Chinese Version of SKT (Syndrom Kurztest): A Short Cognitive Performance Test for the Assessment of Memory and Attention

**DOI:** 10.3390/diagnostics11122253

**Published:** 2021-12-01

**Authors:** Yao Lu, Jingchao Hu, Mark Stemmler, Qihao Guo

**Affiliations:** 1Department of Gerontology, Shanghai Jiao Tong University Affiliated Sixth People’s Hospital, Shanghai 200233, China; luyao1103@126.com; 2School of Nursing, Shanghai Jiao Tong University, Shanghai 200025, China; hujc1126@163.com; 3Department of Psychology, University of Erlangen-Nuremberg, 91052 Erlangen, Germany

**Keywords:** SKT (Syndrom Kurztest), validation, mild cognitive impairment, screening tool

## Abstract

(1) Background: The SKT (Syndrom Kurztest) is a short cognitive performance test that consists of nine subtests and assesses deficits of memory and attention. This study was aimed at exploring the SKT target population in China and evaluating the reliability and validity of the Chinese version of the SKT; (2) Methods: A total of 1624 patients aged over 60 years old were recruited in the Sixth People’s Hospital in Shanghai. The SKT raw scores were recorded. Cronbach’s alpha coefficient was determined to assess the internal consistency reliability of the SKT. Principal factor analysis was performed to evaluate the factor structure of the SKT subtests. Correlation analyses were carried out to confirm the relationship between the modified SKT and standardized neuropsychological tests. The influence of age and educational years on SKT raw scores were detected using multiple regression analyses. Validations of the SKT subtests for detecting Mild Cognitive Impairment (MCI) from Negative Control(NC)(were determined by Receiver operating characteristic (ROC) curves; (3) Results: The internal consistency among the subtests’ scores was high: Cronbach’s α = 0.827. The SKT memory test provided a high predictive validity in detecting aMCI with a sensitivity of 90.1% and specificity of 79.3%. (4) Conclusions: Based on our experience with 1624 elderly patients in Shanghai, the Chinese version of SKT has good stability and may be a reliable and valid screening tool for detecting MCI.

## 1. Introduction

Alzheimer‘s disease (AD) has become one of the most common neurodegenerative diseases that occur with aging. Clinically, it is characterized by cognitive impairment, executive dysfunction, and personality and behavior changes. Among these, cognitive impairment is the core and first symptom of various types of dementia. Different degrees of cognitive impairment will not only affect the health and quality of life of the elderly but also increase the socio-economic burden on families and caregivers. The clinical development process of AD can be divided into preclinical stage, mild cognitive impairment [1] (Mild Cognitive Impairment, MCI), and dementia stage. Amnestic form of MCI (aMCI) is a common subtype of MCI (amnestic, dysexecutive, and mixed) based on impaired performance on both memory measures (Rey AVLT delayed recall and recognition). At present, there are approximately 36.5 million patients with Alzheimer’s disease in the world, and there are more than 8 million patients with Alzheimer’s disease in China [2]. AD has posed a major threat to the health of the elderly. However, due to the lack of specific treatments for AD at present, early intervention in MCI or the preclinical stage is of great significance for delaying the progression of the disease and even reversing the course of the disease. It is of certain importance to use cognitive screening tools before comprehensive neuropsychological assessment. Therefore, the selection of appropriate and highly sensitive screening tools is essential for the early diagnosis of MCI and AD.

Currently, the most frequently used screening tools for AD include the Mini-Mental Status Examination (MMSE), Montreal Cognitive Assessment (MoCA), etc. Since tools such as MMSE are usually affected by education level and age, and the speed of cognitive processing has not been considered, they may not be able to identify subtle memory changes. SKT (Syndrome Kurztest) is an internationally commonly used cognitive screening tool for assessing memory and attention published in Germany [3]. The SKT comprises nine subtests, three of which refer to memory, the remaining six subtests measure speed of information processing. An overview of the subtests and tasks to be completed is given in Table 1 [4]. The SKT, with its duration of only about 10 min, seems to be a reasonable alternative in the context of MCI and dementia when compared to other established tests such as the MoCA [5].

To further verify the applicability and stability of SKT in different languages and cultural backgrounds, it has been translated into 11 different language versions since 1977 and carried out in Spain, England, Chile, South Korea, Brazil, the United States, and other countries [6,7]. Cross-cultural verification shows that the structure of SKT has excellent stability. It is verified by cross-cultural research that SKT is not affected by education level and takes into account the factor of speed of information processing [8]. Although the samples have certain heterogeneity, SKT can still fully detect the decline in memory and attention and is suitable for the early screening of mild cognitive impairment (MCI), dementia, and the severity of cognitive impairment.

China has the largest population of people with dementia. In that sense, we need a screening tool that can be used to assess different degrees of cognitive impairment. At the same time, the applicability needs to be considered, not only its adaptation to the ability of different types of dementia patients but also with regard to the usability for clinicians, general practitioners, nurses, etc. Additionally, the application is relatively easy and the tasks of the SKT are presented in a playful way for the patients. Thus, this study aims to explore the application of the SKT in a target population in China and to assess its validity.

## 2. Materials and Methods

### 2.1. Producing the Chinese Version of the SKT

The Chinese version of the SKT was translated and culturally adapted to the Chinese population. Following the guidelines introduced for the translation and cultural adaptation of the SKT [9], the SKT test materials were modified. First, some of the pictures included in Subtest 1 were changed: sofa to bicycle, pentacle to flower, goblet to teacup, suitcase to umbrella. Substituted pictures were taken from the original SKT test material, which would be familiar to the Chinese elderly. Secondly, the English alphabetic characters in Subtest 7 were changed to Chinese characters: ‘AB’ in Form A to ‘Jia-Yi’.

### 2.2. Participants

The present study included all the 1624 patients (599 cognitively normal controls, 359 subjectively cognitively impaired, 666 individuals with mild cognitive impairment (MCI)) referred between May 2019 to July 2021 to the Sixth People’s Hospital in Shanghai fulfilling the following criteria: (1) age 60 years or older, (2) diagnosis of mild cognitive impairment (MCI, in accordance with the consensus criteria [10]), subjective cognitive impairment (SCD, meeting two major features of SCD criteria proposed by SCD-initiative) [11], (3) complete assessment with all SKT subtests. As an indicator of the clinical severity of SCD and MCI, assignment to stage II (only subjective impairment) or stage III (MCI) of the Global Deterioration Scale (GDS) [12] was required. Exclusion criteria were (1) age below 60 years; (2) all diagnoses other than those required for inclusion, e.g., other forms of dementia (Parkinson’s disease, stroke, encephalitis, meningitis, alcoholism, drug abuse, and head trauma), other forms of depression; and (3) not being able to complete all SKT subtests (e.g., due to severe hearing and vision impairments, due to not being able to understand the test instructions).

Recruited patients further diagnosed with Pre-MCI subjective cognitive decline (SCD) was based on the conceptual framework for the research of SCD [10]. According to this method, patients diagnosed as SCD should meet both standards: (1) Self-experienced persistent decline in cognitive capacity in comparison with a previously normal status and unrelated to an acute event; (2) Normal age-, gender-, and education-adjusted performance on standardized cognitive tests, which are used to classify mild cognitive impairment (MCI) or prodromal AD. What can be explained by a psychiatric or neurologic disease (apart from AD), medical disorder, medication, or substance use should be excluded.

Recruited individuals classified as MCI were based on an actuarial neuropsychological method proposed by Jak and Bondi [11]. According to this method, diagnosis of MCI was given if the participant met one of the following criteria: (1) impaired scores (defined as >1 standard deviation (SD) below the age-corrected normative mean) on two of the six neuropsychological indexes in the same cognitive domain (AVLT 30-min delayed free recall and AVLT recognition for memory, AFT and BNT for language, STT-A and STT-B for executive function); (2) impaired scores (defined as >1 SD below the age-corrected normative mean) in each of the three cognitive domains; (3) Functional Assessment Questionnaire (FAQ) score ≥ 9.

Written informed consent was obtained from all the participants or their caregivers. The ethics committee of Shanghai Jiao Tong University Affiliated Sixth People’s Hospital approved this study. The approval number is 2019-041. The date of review is 25 April 2019. Permission to adapt the Chinese version of SKT was obtained from the original authors.

### 2.3. Neuropsychological Assessment

Besides the SKT, all the participants underwent the standard neuropsychological tests, including Chinse adapted version of MMSE [13], MoCA-B [14], Addenbrooke’s Cognitive Examination III (ACE-III) [15]. Global functional capacity was assessed by the Everyday Cognition (ECOG), the Functional Assessment Questionnaire (FAQ). Part of the participants underwent the Chinese adapted Auditory Verbal Learning Test (AVLT, *n* = 574) [16] and the Brief Visuospatial Memory Test-Revised [17] (BVMT-R, *n* = 520). All the neuropsychological assessments were carried out in Mandarin Chinese by trained raters blind to the diagnosis.

### 2.4. Statistical Analysis

SKT raw scores were entered in a database. Time taken when completing the tests 1, 3, 4, 5, 6, 7 was documented (Maximum: 60 s), and missing or not remembered objects were recorded in test 2, 8, 9 (Maximum: 12). The comparison of demographic characteristics, neuropsychological assessments, and SKT subtest raw scores between NC, SCD, and MCI were conducted using ANOVA and further analyzed by LSD post-hoc analysis. Chi-square tests were applied to categorical data. Cronbach’s alpha coefficient was determined to assess the internal consistency reliability of the SKT. Principal factor analysis was performed to evaluate the factor structure of the SKT subtests. Concurrent validity of SKT with MMSE, MoCA-B was assessed using Linear Regression analysis. Pearson correlation analysis was used to evaluate the correlations between the SKT subtests in memory domain and AVLT, BVMT-R separately. Multiple regression analysis was used to evaluate effects of age and education years on the performance of SKT and other standardized neuropsychological tests. Receiver operating characteristic (ROC) curves were used to determine the ability of SKT memory subtests in discriminating participants with MCI from NC. Statistical analyses were conducted using SPSS Statistics 22.0 (IBM Ltd, Chicago, IL, USA) and MedCalc 19.0 (MedCalc Software Ltd, Ostend, Belgium).

## 3. Results

### 3.1. Demographic Characteristics and Performance of Neuropsychological Tests Subsubsection

The demographics and performance of standardized neuropsychological tests in the groups of NC, SCD, and MCI are shown in Table 2. The study population consisted of 737 males and 887 females whose mean age was 69.29 ± 7.15 years (range 60–89 years) with a mean duration of education of 11.47 ± 3.39 years (range 0–25 years). The MCI group was significantly older than the SCD group and the NC group, with significantly lower education level than the other two groups. In comparison to the NC group and the SCD group, the MCI group showed the worst performance in all the cognitive screening tests (MMSE, MoCA-B, ACE-III, AVLT, BVMT-R). The SCD group scored significantly lower in MMSE and MoCA-B compared to the NC group. The MCI group showed functional decline compared to the NC group assessed by ADL. However, the three groups showed no statistical difference in the ECOG.

### 3.2. Psychometric Properties of the Chinese Version of SKT

The raw scores of SKT of the three groups are compiled together with the results of the group comparison in Table 3. For all of the SKT Subtests, the raw score of each Subtest was negatively correlated with the corresponding MMSE and MoCA-B scores. The Pearson correlation coefficient ranged from −0.258 to −0.621 (*p* < 0.001) for the SKT score and MMSE score and ranged from −0.236 to −0.636 (*p* < 0.001) for the SKT score and MoCA-B score.

The internal consistency among the subtests’ scores was high: Cronbach’s α = 0.827. For the subtests of memory dimension (Subtest 2, 8, 9), the Cronbach’s α was 0.800. As for the attention dimension (Subtest 1, 3, 4, 5, 6, 7, 10), Cronbach’s α was equal to 0.807. To validate the dimensions underlying the SKT, a dimension reduction analysis was carried out on the raw scores of the nine subtests. The rotated component matrix is presented in Table 4. The three memory subtests (Subtests 2, 8, 9), which include immediate and delayed recall plus recognition memory, have uniformly high loadings on factor 2. All of the remaining subtests have significant loadings on factor 1.

Auditory Verbal Learning Test (AVLT) and the Brief Visuospatial Memory Test-Revised (BVMT-R) are both standardized cognitive screening tools examining verbal memory and visual memory, respectively. We carried out the correlation analyses between SKT memory tests and AVLT and BVMT-R (see Table 5). The SKT subtests of memory domain were more closely correlated to the AVLT. The correlated coefficients of immediate recall and delayed recall were −0.519 and −0.603, respectively. Of all the three memory types, delayed memory was the most closely correlated through comparison (subtest 8 × AVLT, *r* = −0.603. subtest 8 × BVMT-R, *r* = −0.513).

### 3.3. Correlations between Demographic Variables and SKT Subtest Scores

We analyzed the effects of age, educational years on the SKT subtest scores, and other cognitive screening tests (see Table 6). Multiple regression analysis revealed that age was positively correlated to SKT raw scores (all *p* < 0.001). Educational years were negatively correlated to the SKT raw score (all *p* < 0.001). Of all the nine subtests, the multiple R value of subtest 2 and subtest 8 were relatively higher than the other subtests (*R* = 0.297, 0.301, respectively). In the other four standardized neuropsychological tests (MMSE, MoCA-B, ACE-III), age and educational years were also correlated to the test scores.

### 3.4. ROC Analyses of Chinese Version of SKT in Identifying MCI from NC

Given that SKT raw scores may be affected by educational years and age, participants in our study were stratified by age and educational years to reduce the demographic bias. Strata of two age levels: young–old (60–74 years), old–old (≥75 years). Strata of different educational levels: lower educational level (educational years ≤ 12 years; *n* = 813, 64.2%), higher educational level (educational years > 12 years; *n* = 452, 35.7%). Among all of the MCI and normal participants, 148 (11.7%) of them had not finished elementary school, and 8 (0.6%) of them were illiterate. Thus, we divided subjects by age and educational years into four groups: group 1 (young–old with lower educational level), group 2 (young–old with higher educational level), group 3 (old–old with lower educational level), and group 4 (old–old with higher educational level).

It has been posited that memory impairment is the protruding symptom of MCI. ROC analyses were performed to evaluate the validation of the SKT subtest of memory dimension (Immediate recall + Delayed recall + Recognition) in discriminating individuals with MCI from NC in different subgroups (Figure 1). In two age strata (young–old and old–old), ROC curve analyses suggested that the SKT memory test can better discriminate MCI from NC in subgroups with a higher educational level (see Table 7). Among the four groups, group 4 had the largest area under the ROC curve (AUC), 0.821 (95%CI: 0.747, 0.876, with sensitivity 84.3% and specificity 62.1%), which is significantly higher than group 1, group 2, and group 3 (*p* = 0.019, 0.035, 0.025, respectively). The AUC showed a significant difference between group 1 and group 2 (AUC = 0.707, 0.767, respectively, *p* = 0.017). The cut-off value (missing or not remembered objects) of group 2 and group 4, who were well educated, was the lowest.

We also carried out the ROC analyses to verify the accuracy of SKT memory tests in discriminating aMCI (amnestic MCI) from NC (see Figure 2). The AUC was 0.878 (95% CI: 0.845, 0.911), with sensitivity 90.10% and specificity 79.3%, indicating that SKT memory test had good accuracy in distinguishing aMCI from NC. ROC analyses of delayed recall also showed good power in detecting aMCI with AUC of 0.827 (95% CI: 0.789, 0.865).

## 4. Discussion

This study supports the notion of the reliability of the Chinese version of SKT. The internal consistency, which is considered an essential aspect of the test battery, was found to be good in the study. The value of Cronbach’s coefficient obtained, α = 0.827, indicates high intercorrelations among subtests. The Cronbach’s coefficient is α = 0.800 for the memory domain and α = 0.807 for the attention domain. The early studies have verified the reliability of SKT in different cultures, which was similar to our research, the Cronbach’s coefficient varies between 0.80 to 0.88 [18,19]. Erzigkeit’s study [20], based on a sample of 3789 subjects, showed that the reliability of memory and attention domain was very good. The Cronbach α value of the memory domain was α = 0.86, and that of the attention domain was α = 0.90.

The factor analysis confirmed the existence of the two domains underlying the SKT subtests (see Table 4). Factor 1 was related to subtests accomplished under time pressure (subtests 1, 3, 4, 5, 6, and 7). Factor 2, which related to subtest 2, 8, and 9, represented cognitive domains of memory. In that sense, those subtests are dependent on attention and memory separately. The results of the factor analysis were similar to the early studies in Germany, the USA, and Korea [21,22,23]. In producing the Chinese version of SKT, many pictures of the original version unfamiliar to the Chinese elderly were replaced. In that sense, we can avoid the situation that the naming may be related to the unfamiliarity of the presented objects and not to attention or memory; that is, if an object cannot be named, it cannot be recalled, thus affecting the scores of the memory domain. Overall, the Chinese version of the SKT kept psychometric properties and factor structure comparable to the original test version.

Our study verified the validation of the Chinese version of SKT in detecting the earlier stage of cognitive impairment stage. We found a substantial correlation between the SKT scores and the MMSE and MoCA-B scores, respectively. For all of the SKT subtests, the raw scores correlated negatively with the corresponding MMSE scores and MoCA-B scores (see Table 3). Between the three subgroups, there were significant statistical differences in all of the subtests of SKT (*p* < 0.001). Following further analysis of the results performed by comparing each subgroup, SKT sub-scores significantly differed between the NC, SCD, and MCI groups. Despite the SCD referring to the self-perception of cognitive performance, which is conceptually independent of the performance of the cognitive test, the SKT subtests showed good sensitivity in identifying the different stages of cognitive impairments. The first symptoms of AD are not limited to memory decline, and patients may report memory decline when they experience declines in other cognitive domains. SKT used questions specifically related to memory functioning (immediate recall, delayed recall, recognition memory), which have strong evidence for an association with preclinical AD. Moreover, the SKT was developed and elaborated using tasks in other cognitive domains beyond the assessment of memory decline (attention). In sum, the SKT may be an excellent predictor of objective performance and may facilitate the identification of the very early decline related to AD.

Very few studies have illustrated the diagnostic power in screening MCI from NC. At first, we analyzed the effects of age and educational years on SKT raw scores. The multiple regression analysis showed that the patients with more educational years and younger age, scored lower on the SKT subtests (see Table 6). Participants in our study were stratified by age and educational years to reduce the demographic bias. Then, ROC analyses were carried out among the four groups. Group 4 had the largest area under the ROC curve (AUC) for the memory test, 0.821 (95% CI: 0.747, 0.876), with sensitivity 84.3% and specificity 62.1%, which is significantly higher than group 1, group 2, and group 3 (*p* = 0.019, 0.035, 0.025, respectively). The AUC showed a significant difference between group 1 and group 2 (AUC = 0.707, 0.767, respectively; *p* = 0.017). The cut-off values of group 2 and group 4 who were well educated were the lowest, suggesting that the cut-off values should be adjusted by educational level. We also noticed that people with lower education yielded lower specificity values, suggesting that mild deficits associated with lower education may be incorrectly identified as having cognitive decline. Overall, based on our experience of using the SKT in Shanghai, age and educational level may affect the accuracy in detecting MCI.

Although in partial disagreement with the original guidelines provided by the Erzigkeit study, that the SKT is an instrument not affected by education [24], our result is consistent with several previous studies that showed that educational levels were associated with the performance of SKT [25,26]. Ostrosky conducted a study on 335 subjects that verified the influence of educational levels on the accuracy of the SKT (97 mild dementia patients and 238 normal people). The results showed that SKT showed sufficient performance for subjects with intermediate to higher education levels, sensitivity (80.5%) and specificity (80.3%); however, for low-educated and illiterate subjects, the sensitivity and specificity of the SKT were significantly lower (75% and 56.7%, respectively). In sum, our comprehensive analyses provided the evidence, in the Chinese population, that SKT scores may be affected by education and age.

Amnestic MCI is characterized by mild impaired performance on memory measures. Thus, we used ROC analyses to evaluate the accuracy of memory tests in SKT to detect aMCI. In our study, 198 subjects were identified as aMCI using the AVLT test. The sensitivity of SKT memory tests for aMCI was 90.1% which provided a high predictive validity in terms of the onset of amnestic MCI, despite age and educational level. Our preliminary exploration indicated that the SKT memory tests may have a very good power to discriminate cases of aMCI from NC.

Of all the nine subtests, subtest 8 (delayed recall) showed a good power to differentiate aMCI from NC (Figure 2). The AUC for the delayed recall was 0.827 (95% Cl 0.789–0.865), with sensitivity 88.8% and specificity 67.4%. The correlation analyses between the SKT memory tests and AVLT and BVMT-R indicated that the memory impairment detected by SKT was embedded with both verbal and visual memory properties (see Table 5). The close correlation of SKT with AVLT demonstrated that SKT memory tests exhibited significant ability to detect verbal memory deficits. The former study of Guo et al. [16] indicated that AVLT is superior to the visual memory test in the stability of diagnoses and prediction of cognitive decline with optimal sensitivity and specificity. The MCI patients exhibit the episodic memory impairment due to pathological changes in the entorhinal cortex and the hippocampus area with preserved semantic function on the distributed cortex. Additionally, Chinese is a logographic language. Material presented visually might be easier to store through double encoding as imaging and linguistic [27]. In that sense, Chinese elders might differ from western elders due to the language differences, and verbal memory screening tools may be more sensitive to detecting memory deficits.

Several limitations should be noted. This study is monocentric, and the participants are mainly from the urban area from Shanghai, thus, we should be aware of the geographic and culture variability across China.

## 5. Conclusions

Overall, our study verified the applicability of the Chinese version of the SKT. Based on our experience with 1624 elderly patients in Shanghai, the Chinese version of the SKT has good stability and, thus, may be a reliable and valid screening tool for detecting MCI. However, test results must be interpreted with caution considering individuals’ age and educational level.

## Figures and Tables

**Figure 1 diagnostics-11-02253-f001:**
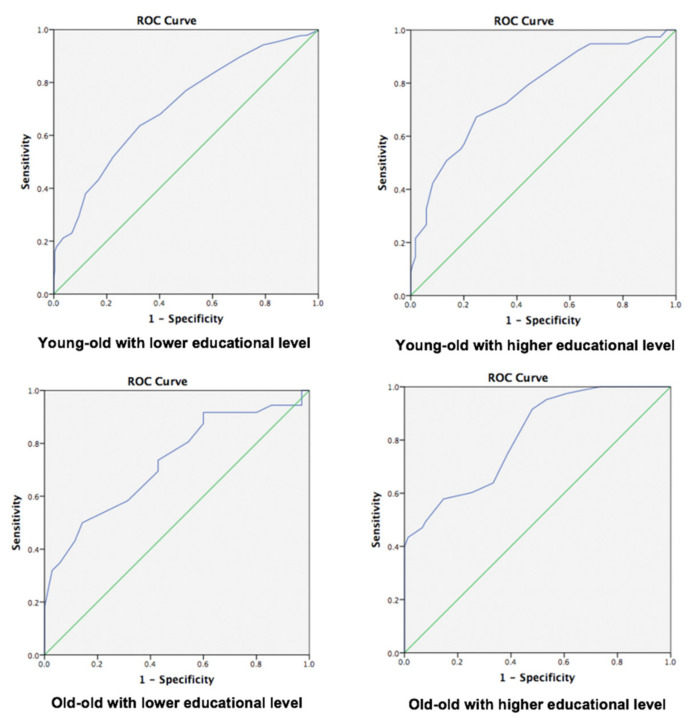
Receiver operating characteristic (ROC) curves of SKT memory tests in detecting MCI patients from NC of different subgroups.

**Figure 2 diagnostics-11-02253-f002:**
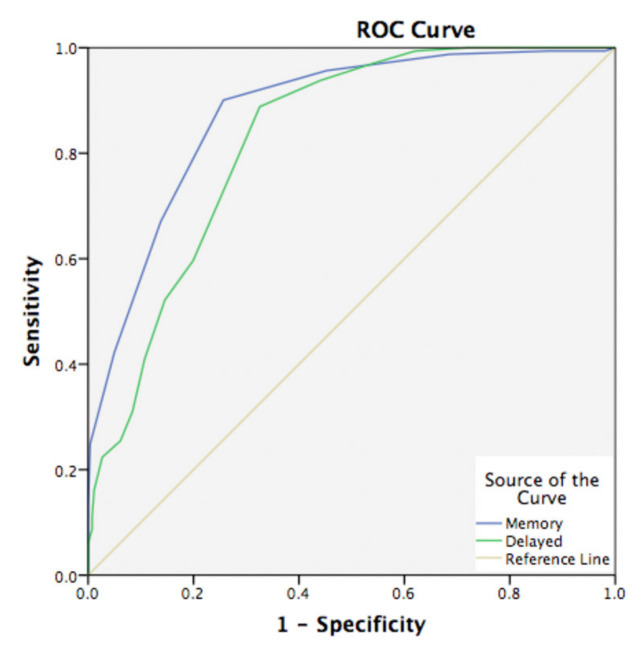
Receiver operating characteristic curves of SKT memory tests in detecting aMCI patients from NC.

**Table 1 diagnostics-11-02253-t001:** Overview of the SKT subtests.

Subtest	Name	Content	Domain
Subtest 1	Naming Objects	12 objects have to be named and memorized at the same time	Attention/Speed
Subtest 2	Immediate Recall	Recall of the objects shown in test 1	Memory
Subtest 3	Naming Numerals	2-digit numbers written on magnetic blocks have to be read out loud	Attention/Speed
Subtest 4	Arranging Blocks	The magnetic blocks have to be arranged in ascending order	Attention/Speed
Subtest 5	Replacing Blocks	The blocks have to be replaced in their original place	Attention/Speed
Subtest 6	Counting Symbols	Target Symbols have to be counted among other distractor symbols	Attention/Speed
Subtest 7	Reversal Naming	Chinese characters are to be read out interchanged, which creates an interference	Attention/Speed
Subtest 8	Delayed Recall	Recall of the objects shown in test 1	Memory
Subtest 9	Recognition	Identification of the objects of test 1 from the table containing 48 objects	Memory

**Table 2 diagnostics-11-02253-t002:** Demographic characteristics and performance of neuropsychological tests.

	NC (*n* = 599)	SCD (*n* = 359)	MCI (*n* = 666)
Age (y)	68.23 ± 5.858	64.51 ± 7.833	70.70 ± 6.858
Sex (M/F)	330/269	116/243	291/375
Years of Education	11.73 ± 3.39	11.69 ± 3.05	11.06 ± 3.56
MMSE	28.07 ± 3.12	27.53 ± 1.81	26.28 ± 1.91
MoCA-B	24.21 ± 4.10	23.99 ± 3.57	23.05 ± 3.93
ACE-III	78.71 ± 9.90	76.41 ± 11.64	75.55 ± 10.31
AVLT (*n* = 574)			
AVLT Immediate recall	18.45 ± 4.96	\	12.24 ± 5.43
AVLT Delay recall	8.96 ± 4.78	\	4.39 ± 3.66
AVLT Recognition	27.63 ± 5.03	\	19.89 ± 5.56
BVMT-R (*n* = 520)			
BVMT-R Immediate recall	21.86 ± 5.72	\	17.28 ± 9.06
BVMT-R Delay recall	8.97 ± 3.76	\	7.48 ± 3.78
BVMT-R Recognition	11.61 ± 1.56	\	9.96 ± 1.95
ECOG	20.43 ± 7.51	19.95.26 ± 6.69	20.73 ± 8.13
ADL	22.87 ± 8.15	23.10 ± 5.57	24.18 ± 8.62

MMSE, Mini-Mental Status Examination; MoCA-B, Montreal Cognitive Assessment; ACE-III, Addenbrooke’s Cognitive Examination III; AVLT, Auditory Verbal Learning Test; BVMT-R, Brief Visuospatial Memory Test-Revised; ECOG, Everyday Cognition; FAQ, Functional Assessment Questionnaire; ADL, Activities of Daily Living; NC, normal control; SCD, subjective cognitive decline; MCI, mild cognitive impairment.

**Table 3 diagnostics-11-02253-t003:** Raw scores for SKT subtests and results of group comparison with MMSE, MoCA-B.

Diagnosis	NC	SCD	MCI	Correlation with MMSE (Beta)	Correlation With MoCA-B (Beta)
Subtest 1	20.74 ± 8.31	20.93 ± 7.13 ^b^	24.11 ± 9.63 ^c^	−0.258 *	−0.236 *
Subtest 2	7.95 ± 3.91	8.57 ± 4.09 ^b^	9.14 ± 3.82 ^c^	−0.489 *	−0.515 *
Subtest 3	23.48 ± 8.09	26.49 ± 17.09 ^a,b^	29.09 ± 11.64 ^c^	−0.344 *	−0.380 *
Subtest 4	17.35 ± 6.69	18.89 ± 7.94 ^a,b^	21.27 ± 9.54 ^c^	−0.540 *	−0.583 *
Subtest 5	23.27 ± 6.69	25.82 ± 8.36 ^a^	25.84 ± 7.60 ^c^	−0.563 *	−0.590 *
Subtest 6	26.45 ± 7.25	29.57 ± 8.02 ^a^	30.73 ± 9.02 ^c^	−0.431 *	−0.470 *
Subtest 7	6.31 ± 2.91	6.60 ± 2.05 ^a,b^	7.47 ± 4.27 ^c^	−0.260 *	−0.276 *
Subtest 8	4.88 ± 3.10	5.73 ± 2.54 ^a,b^	6.96 ± 2.77 ^c^	−0.588 *	−0.636 *
Subtest 9	1.87 ± 3.13	2.13 ± 1.99 ^a,b^	2.99 ± 2.39 ^c^	−0.621 *	−0.609 *

NC, normal control; SCD, subjective cognitive decline; MCI, mild cognitive impairment. ^a^: SCD compared to NC, *p* < 0.001. ^b^: SCD compared to MCI, *p* < 0.001. ^c^: MCI compared to NC, *p* < 0.001. * *p* < 0.001.

**Table 4 diagnostics-11-02253-t004:** Rotated Component Matrix.

Subtest	Component	
	Factor 1	Factor 2
Replacing blocks (s)	0.779	
Counting symbols (s)	0.762	
Naming numerals (s)	0.738	
Arranging blocks (s)	0.703	
Naming objects (s)	0.572	
Reversal naming (s)	0.581	
Delayed recall		0.870
Immediate recall		0.863
Recognition		0.833

**Table 5 diagnostics-11-02253-t005:** Correlation analysis of SKT subtests of memory domain and standardized neuropsychological tests.

	AVLT (Standardized Coefficients)			BVMT-R (Standardized Coefficient)		
	Immediate Recall	Delayed Recall	Recognition	Immediate Recall	Delayed Recall	Recognition
SKT Subtest 2 (Immediate recall)	−0.511 ^a^	\	\	−0.306 ^a^	\	\
SKT Subtest 8 (Delayed recall)	\	−0.603 ^a^	\	\	−0.513 ^a^	\
SKT Subtest 9 (Recognition)	\	\	−0.231 ^a^	\	\	−0.033

AVLT, Auditory Verbal Learning Test; BVMT-R, Brief Visuospatial Memory Test-Revised. ^a^: *p* < 0.001.

**Table 6 diagnostics-11-02253-t006:** Multiple regression analysis of demographic variables and neuropsychological assessments.

	Constant	Age(Coefficients, Beta)	Edu-Years(Coefficients, Beta))	*R*
SKT				
Subtest 1	7.64	0.256 ^a^	−0.274 ^a^	0.184
Subtest 2	−0.13	0.111 ^a^	−0.069 ^a^	0.297
Subtest 3	4.41	0.088 ^a^	−0.167 ^a^	0.172
Subtest 4	0.06	0.531 ^a^	−0.499 ^a^	0.151
Subtest 5	−1.19	0.299 ^a^	−0.418 ^a^	0.203
Subtest 6	2.19	0.182 ^a^	−0.324 ^a^	0.229
Subtest 7	18.29	0.250 ^a^	−0.580 ^a^	0.157
Subtest 8	1.62	0.091 ^a^	−0.088 ^a^	0.301
Subtest 9	−2.61	0.080 ^a^	−0.074 ^a^	0.236
MMSE	30.12	−0.075 ^a^	0.184 ^a^	0.303
MoCA-B	30.94	−0.178 ^a^	0.408 ^a^	0.392
ACE-III	83.11	−0.347 ^a^	1.516 ^a^	0.492

MMSE, Mini-Mental Status Examination; MoCA-B, Montreal Cognitive Assessment; ACE-III, Addenbrooke’s Cognitive Examination III. ^a^: *p* < 0.001.

**Table 7 diagnostics-11-02253-t007:** ROC analyses of SKT memory test in detecting MCI patients from NC of different subgroups.

	AUC	Sensitivity	Specificity	Cut-off
NC✖️MCI (Delay Recall + Immediate Recall + Recognition) (MCI *n* = 666)				
Group 1 Young–old, Lower Edu (*n* = 700)	0.707 (95%CI 0.669, 0.746)	63.6%	67.5%	14.5
Group 2 Young–old, higher Edu (*n* = 289)	0.767 (95%CI 0.711, 0.823)	67.2%	75.3%	13.5
Group 3 Old–old, Lower Edu (*n* = 113)	0.727 (95%CI 0.631, 0.824)	73.6%	57.1%	16.5
Group 4 Old–old, Higher Edu (*n* = 163)	0.821 (95%CI 0.747, 0.876)	84.3%	62.1%	13.5
NC✖️aMCI (Delay Recall + Immediate Recall + Recognition) (aMCI *n* = 198)				
	0.878 (95%CI 0.845, 0.911)	90.1%	79.3%	13.5
	NC✖️aMCI (Delay Recall) (aMCI *n* = 198)			
	0.827 (95%CI 0.789, 0.865)	88.8%	67.4%	5.5

NC, normal control; SCD, subjective cognitive decline; MCI, mild cognitive impairment; aMCI, amnestic mild cognitive impairment.

## Data Availability

All data generated or analyzed during this study are included in this published article.

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
