# Peer review of "Validation of Chinese Version of SKT (Syndrom Kurztest): A Short Cognitive Performance Test for the Assessment of Memory and Attention"

_diagnostics, 2021, doi:10.3390/diagnostics11122253_

Round 1
Reviewer 1 Report
This manuscript reports on a study that examined the validiy and reliabilitythe Chinese version of SKT. The conducted study is well designed and collected a sufficient number of participants to reveal its power as a screening tool. I have no disagreement with authors' conclusion, and the SKT is considered to have an acceptable performance as a screening test (although I personally do not feel it to be particularly superior to the MoCA or MMSE).
The following comments are all minor points;
In Table 2, it is not informative to explain statistically significant differences in each group, because participants in each groups are so big that subtle difference can be deemed as "statistically signiricant." So you may want to delete all p-values. You may want to briefly describe the trend of this population by qualitatively, or compare it with studies in other countries and with populations using other assessment scales.
Similarly, the p-values in Table 3 do not take multiplicity into account, so it seems less significant to explain the results. Throughout this manuscript, I saw tons of unnecessary p-values so I suggest authors limit p-values as much as possible and/or clearly state analyses as exploratory.
Author Response
Dear Hoshi Liu and reviewer, Submission ID diagnostics-1406318 “Validation of Chinese version of SKT (Syndrom Kurztest (SKT): a short cognitive performance screening test for the assessment of memory and attention.” Thank you so much for your constructive feedback and comments on our manuscript. Those comments are all valuable and very helpful for revising and improving our paper, as well as the important guiding significance to our researchers. We have studied these comments carefully and have made corrections which we hope to meet with approval. All changes within the revised manuscript are in 'track changes' mode. The main corrections in the paper and the point-by-point responses to the reviewers’ comments are as follow: Point-by point Response to Reviewers Reviewer #1 Comments 1: In Table 2, it is not informative to explain statistically significant differences in each group, because participants in each group are so big that subtle difference can be deemed as "statistically significant." So, you may want to delete all p-values. You may want to briefly describe the trend of this population by qualitatively or compare it with studies in other countries and with populations using other assessment scales. Author Response: Thanks for your constructive suggestions. We have deleted the p-values in Table 2. The demographic information was qualifiedly shown in the table briefly. Please see the changes on Page 5, Table 2. Comments 2: Similarly, the p-values in Table 3 do not take multiplicity into account, so it seems less significant to explain the results. Throughout this manuscript, I saw tons of unnecessary p-values, so I suggest authors limit p-values as much as possible and/or state analyses as exploratory. Author Response: Thanks for your comment. Your constructive suggestions helped us to simplify the table. We have already deleted the p-value row in table 3, instead of using the subscripts to show the statistical differences. Please see the changes on Page 5, table 3. And also, we have checked other tables making sure to limit the use of p-values to ensure the data in the forms is more streamlined. Thanks again for your suggestions. Sincerely yours, Yao Lu Department of gerontology, Shanghai Jiao Tong university-affiliated sixth people’s hospital, Shanghai, China, 200233. Tel: +86 13162995901 e-mail: [email protected]
Reviewer 2 Report
I wanted to congratulate the authors, this article is a new diagnostic tool. The Kurtz test Represents a new diagnostic tool in the Chinese sample.
I need to suggest minor changes, the first one is related with the ethics committee, it is necessary to write the act number and date. It is important to clarify if was obtained a permission for the original authors to use the Kurtz test in the Chinese population. Is very important to clarify everything related with copyright of this test.
Second. The results demonstrate a huge selection bias related with a mono centric study, however I wanted to know how the research team ameliorate the effects of selection bias.
It is important to analyze how many patients have no formal education, or classified as illiterate.
When I read the text is not clear the cut off point for mci and subjective memory complaint.
I suggest to clarify the exclusion and inclusion criteria, and how the protocol can differentiate between Dementia and mci.
It is important to clarify the protocol to diagnose mci (Petersen s criteria or DSM v) and if was performed a work up of reversible causes related with cognitive impairment
Author Response
Submission ID diagnostics-1406318
“Validation of Chinese version of SKT (Syndrom Kurztest (SKT): a short cognitive performance screening test for the assessment of memory and attention.”
Thank you so much for your constructive feedback and comments on our manuscript. Those comments are all valuable and very helpful for revising and improving our paper, as well as the important guiding significance to our researchers. We have studied these comments carefully and have made corrections which we hope to meet with approval. All changes within the revised manuscript are in 'track changes' mode. The main corrections in the paper and the point-by-point responses to the reviewers’ comments are as follow:
Point-by point Response to Reviewers
Reviewer #2
Comments 1: I need to suggest minor changes, the first one is related to the ethics committee, it is necessary to write the act number and date. It is important to clarify if was obtained permission for the original authors to use the Kurtz test in the Chinese population. Is very important to clarify everything related to copyright of this test.
Author Response: Thanks for your constructive comment. The present study included all the 1624 patients (599 cognitively normal controls, 359 subjectively cognitively impaired, 666 individuals with mild cognitive impairment (MCI)) referred between May 2019 to July 2021 to the Sixth People’s Hospital in Shanghai. According to the approval letter of ethics committee of Shanghai Sixth People’s Hospital, the approval number is 2019-041. The date of review is April 25 in 2019. The approval letter is attached below. Please see changes on Page 8, lines 227-229; Page 17, lines 456-458.
Permission to adapt the Chinese version of SKT had been obtained from the original authors. To act the international validation in China, we designed our study following the proposal provided by Prof. Stemmler in German, who is the copyright holder. And we have clarified this part in the article. Please see changes on Page 8, lines 227-229; Page 17, lines 456-458.
Comments 2: The results demonstrate a huge selection bias related to a mono-centric study, however, I wanted to know how the research team ameliorate the effects of selection bias.
Author Response: Thanks for your comments. Since our research is monocentric, there are certain limitations in participants' selection regarding geographic and cultural characteristics. We have tried to lower the influence of selection bias through the following ways. First of all, the establishment of our database for clinical research on pre-dementia diagnosis mainly comes from various communities and is not limited to hospitals, so that the age, gender, occupation, cultural background, and general health status of participants could be relatively randomized. Participants are informed of our research through neighborhood committees, news, self-media, and volunteer to participate in our research. And the communities we cooperate with radiate multiple districts in Shanghai, including Xuhui District, Pudong New District, Yangpu District (suburban district), Jing'an District, Huangpu District, Qingpu District (suburban district). Second, we try to collect as much neuropsychological test data as possible to reduce selection bias. Our clinical database has been established in 2018, and the data of neuropsychologic tests of more than 3,000 patients have been collected. In this study, 1624 participants have conducted the SKT. Although our database is monocentric, we try to collect more data both horizontally and longitudinally.
The selection bias you mentioned is indeed the limitation of our research . We hope that we can further expand the scope of our research and further explore the application experience in other regions of China, which will be the next step.
Comments 3: It is important to analyze how many patients have no formal education or are classified as illiterate.
Author Response: Thanks for your comments. We divided the participants into two educational levels according to educational years. Lower educational level (educational years <=12years; N=813,64.2%) referred to below the junior high school education level. Higher educational level (educational years >12years; N=452,35.7%) referred to high school education level or above. We have also analyzed the number of patients who have no formal education or were illiterate. 148(11.7%) participants hadn’t finished elementary school, and 8(0.6%) participants were illiterate. Please see changes on Page 12, lines 314-318.
Comments 4: When I read the text is not clear the cut off point for mci and subjective memory complaint.
Author Response: Thanks for your comments. In our study, we have also conducted the ROC analyses of the SKT memory test in distinguishing the MCI patients and SCD patients. The subtests of the memory domain showed limited sensitivity and specificity in differentiating the MCI and SCD. Please see the results in the table and figure below. If you find this part of the analysis should also display in our article, we could put this part of the results in the supplementary material.
AUC |
Sensitivity |
Specificity |
Cut-off |
|
SCD✖️MCI (Delay Recall+ Immediate Recall+ Recognition) (MCI n=666,SCD=359) |
||||
|
0.651 (95%CI0.616, 0.686) |
57.0% |
62.1% |
15.5 |
SCD✖️MCI (Delay Recall) (MCI n=666, SCD=359) |
||||
|
0.640 (95%CI0.605, 0.674) |
50.5% |
68.8% |
7.5 |
Comments 5: I suggest clarifying the exclusion and inclusion criteria, and how the protocol can differentiate between Dementia and MCI.
Author Response: Thanks for your suggestions. We have further illustrated the inclusion and exclusion criteria in our article.
Recruited Individuals classified as MCI were based on an actuarial neuropsychological method proposed by Jak and Bondi [1]. According to this method, diagnosis of MCI was given if the participant met one of the following criteria: (1) impaired scores (defined as >1 standard deviation (SD) below the age-corrected normative mean) on two of the six neuropsychological indexes in the same cognitive domain (AVLT 30-minute delayed free recall and AVLT recognition for memory, AFT and BNT for language, STT-A, and STT-B for executive function); (2) impaired scores (defined as >1 SD below the age-corrected normative mean) in each of the three cognitive domains; (3) Functional Assessment Questionnaire (FAQ) score≥9.
The diagnosis of Pre-MCI subjective cognitive decline (SCD) was based on the conceptual framework for the research of SCD[2]. According to this method, patients diagnosed with SCD should meet both two standards: (1) Self-experienced persistent decline in cognitive capacity in comparison with a previously normal status and unrelated to an acute event. (2) Normal age-, gender-, and education-adjusted performance on standardized cognitive tests, which are used to classify mild cognitive impairment (MCI) or prodromal AD. What can be explained by a psychiatric or neurologic disease (apart from AD), medical disorder, medication, or substance use should be excluded.
Please see the changes on Page 8, lines 207-224.
Sincerely yours,
Yao Lu
Department of gerontology, Shanghai Jiao Tong university-affiliated sixth people’s hospital, Shanghai, China, 200233.
Tel: +86 13162995901
e-mail: [email protected]
References
[1]. Bondi MW, Edmonds EC, Jak AJ, Clark LR, Delano-Wood L, McDonald CR, et al. Neuropsychological criteria for mild cog-nitive impairment improves diagnostic precision, biomarker associations, and progression rates. Journal of Alzheimer's disease : JAD. 2014;42(1):275-89. Epub 2014/05/23. doi: 10.3233/jad-140276. PubMed PMID: 24844687; PubMed Central PMCID: PMCPMC4133291.
[2]. Jessen F, Amariglio RE, van Boxtel M, Breteler M, Ceccaldi M, Chételat G, et al. A conceptual framework for research on sub-jective cognitive decline in preclinical Alzheimer's disease. Alzheimer's & dementia : the journal of the Alzheimer's Associa-tion. 2014;10(6):844-52. Epub 2014/05/07. doi: 10.1016/j.jalz.2014.01.001. PubMed PMID: 24798886; PubMed Central PMCID: PMCPMC4317324.
Round 2
Reviewer 2 Report
All the corrections requested by the reviewer were made, i suggest to accept the article in the current form